# Non-Contact Thermometer for Improved Air Temperature Measurements

**DOI:** 10.3390/s23041908

**Published:** 2023-02-08

**Authors:** Marco Pisani, Milena Astrua, Andrea Merlone

**Affiliations:** Istituto Nazionale di Ricerca Metrologica (INRIM), Strada delle Cacce 91, 10135 Torino, Italy

**Keywords:** air temperature, speed of sound, metrology, meteorology

## Abstract

A compact thermometer for air temperature based on the measurement of the speed of sound was developed at INRIM. This paper focuses on the comparison of this instrument with platinum resistance thermometers in a climatic chamber over a temperature range (−30 ÷ +55) °C, relative humidity (10 ÷ 90)%rh, and irradiation (>1 kW/m^2^) values similar to those of surface atmospheric conditions. Overall uncertainty values of 0.2 °C over the range from −30 °C to +30 °C, and from 0.6 °C to +55 °C, were found. Moreover, the instrument proved to be immune to irradiation errors and free from the need for temperature calibration.

## 1. Introduction

Air temperature is measured for a multitude of purposes. Atmospheric air temperature is the key variable in indoor air conditioning; laboratory control, where it is critical for precision in dimensional and mass measurements; in industrial processes of refrigeration; and, most importantly of all, in meteorological observations and climate studies. Due to the importance of this variable and its role of influencing factors for a large portion of human activities, air temperature can be considered as the most measured quantity in life, industry, technology, and science. Despite its importance, the accuracy of air temperature data and lack of complete understanding of uncertainty in measurements seems to be a widely underestimated problem. Understanding and fully evaluating measurement uncertainty for air temperature measurements is an open scientific and technical issue motivating research efforts and discussion of the CIPM (International Committee for Weights and Measures) CCT (Consultative Committee for Thermometry) [1], where a specific task group was formed in 2021; in WMO (World Meteorological Organization) expert teams; and of the GCOS (Global Climate Observing System).

Measurements of air temperature at present are almost exclusively based on contact thermometers: instruments equipped with a sensor which requires physical contact with the measurand, in this case the air. These are mainly resistance thermometers (thermistors and platinum resistance thermometers (PRT)), thermocouples, and the many liquid-in-glass thermometers still present in older installations. PRTs are the most accurate temperature sensors used in metrology and which are also used as standard instruments (SPRTs) in the realization and dissemination of the International Temperature Scale. The highest-quality meteorological and industrial thermometers are also PRTs. The main limits of PRTs are due to self-heating and the fact that these are contact systems, and thus affected by the radiative effects studied in [2,3]. In particular, the work of [3] demonstrated that radiative errors increase with the sensor diameter. Due to their physical construction, frequently involving metal sheets, PRTs are also affected by significant thermal inertia responsible for their time constant, which frequently does not fulfil requirements in terms of response time [4]. Vice versa, the greater the sensor diameter, the smaller the effect of self-heating, and several construction compromises have been adopted by manufacturers in order to minimize both, while using a sufficient amount of feeding current in order to achieve sensitivity. Due to their small size and higher sensitivity in terms of changes in their resistance to temperature, thermistors are less affected by radiative errors with respect to PRTs, but are highly limited by the effects of self-heating, reducing overall accuracy and non-linearity in their sensitivity curves, and causing errors in their calibration interpolation curves [5]. Furthermore, thermistors are based on the conductibility of a semiconductor, which is strongly determined by the manufacturing process and prone to important drifts along their lifetime; thus, the intrinsic accuracy is low and frequent calibrations are needed. Thermocouples are a possible alternative, since they can be as small as fractions of millimetres, are free from self-heating processes, and are based on the well-defined physical principle of the contact potential between different metal alloys; nevertheless, the intrinsic accuracy is also not sufficient for use in demanding applications. One of the latest examples of the use of thermocouples for air temperature measurements can be found in [6].

Silicon diodes and capacitive sensors are also used for lower-accuracy systems. In meteorology, other systems are involved in generating atmospheric air temperature “data products”, such as LIDARs or microwave temperature profilers [7] and remote sensing, the measurement results of which are compared and validated with radio-sounding and ground-based stations involving contact sensors; however, these methods are only suitable for measurements over long distances and cannot be used for local measurements. Optical measurements of air temperature are also performed, mainly devoted to the determination of the air refractive index in order to improve interferometric measurements [8]. For the same purpose, the research field on the measurement of air temperature based on the propagation of acoustic waves in air is growing [9,10]. A review of the existing techniques can be found in [11].

Regardless of the above, laboratory room temperature control, indoor air conditioning, and all ground-based meteorological and climatological stations involve contact thermometers. In order to limit errors caused by radiative sources, air temperature sensors are often surrounded by shields or screens. These enclosures are available in several different designs; therefore, in 2007, an ISO standard was prepared to define the main relevant characteristics of these shields [12]. Other sensor-related influences include dew condensation; evaporative cooling due to phase transitions; direct, diffuse, and reflected solar radiation; and calibration procedures. All of these become significant factors of influence and uncertainty when thermometers are used in the field for environmental observations. Completing a thermodynamic model and analytical function for the description of the interaction between a real gas and a real sensor in air temperature measurement is, at present, almost impossible, given the complexity and mutual interaction of the many factors influencing the thermodynamic equilibrium between the gas and the probe. In addition, further contributions to overall measurement uncertainty arise from the physical and technical features of the sensor, including its surface properties, dimensions, and coupling with protective shields, when these are present; these are also numerous and diverse for the different technological solutions adopted to increase accuracy.

A number of experiments and studies have already investigated specific aspects of atmospheric air temperature measurements, quantities of influence, and uncertainty components. During the MeteoMet projects [13,14] some of these effects were evaluated mainly in terms of single components in the overall measurement uncertainty budget: sensor dynamics, direct and reflected radiation [15,16], self-heating of meteorological sensors [17], and wind and rain.

Not only is the uncertainty of in situ air temperature measurement presently a subject of study, but the calibration of thermometers in air still requires definitions of procedures and prescriptions, including evaluation of calibration uncertainty, which is a significant component of the overall measurement uncertainty. In the absence of standardized procedures for the calibration of thermometers in air, sensors are usually calibrated in a liquid bath under adiabatic conditions, while any measurement of air temperature takes place in anything but adiabatic conditions. When the thermometers are calibrated in liquid baths, the errors due to the different measurement conditions, including self-heating dissipation, condensation, and radiation, could be underestimated. This underestimation can be worse when higher electrical currents are applied to the thermometer by readout systems and data loggers to achieve higher sensitivity.

In the present work, we describe a compact acoustic thermometer (CAT), based on the measurement of the speed of sound, that proved to be immune from radiative errors and able to potentially reduce other effects, such as condensation and convection. A further advantage of this device is that traceability to the SI units is guaranteed by mechanical quantities: therefore, it does not require calibration against temperature reference values. The working principle and the practical realization of the CAT are described. An experimental campaign aimed at comparing the reading of the CAT against classical platinum thermometry in a wide interval of temperature, humidity, and irradiation conditions was carried out, and the results presented and discussed.

## 2. The Working Principle

The CAT is based on an accurate measurement of the speed of sound, which extensively depends upon air temperature [18]. The principle has been already exploited in an experiment in the frame of a European research project [19] with the aim of improving the knowledge of the speed of sound in air. In the CAT, an acoustic wave at an arbitrary fixed frequency ν is generated by a transducer driven by a synthesizer. The wave travels in air for a given space and reaches a sound pressure sensor placed at an arbitrary distance *L*, where it is converted into an electric signal with frequency *ν*. The phase *φ* between the electrical signal at the generator and the electrical signal at the receiver is measured. The mathematical relation between the phase φ and the acoustic wavelength *λ* is the following:(1)$$\varphi =\frac{L}{\lambda }\;2\pi$$

Since the speed of sound *u = λν*, Equation (1) can also be written as
(2)$$\varphi =\frac{\nu L}{u}\;2\pi$$

At constant environmental parameters, the phase *φ* is constant. When the distance *L* between the transmitter and the receiver is changed with a known motion law (preferably at a constant speed), the recorded phase changes accordingly:(3)$$\frac{d\varphi }{dL}=\frac{\nu }{u}\;2\pi$$

It is easy to see that the phase changes by a complete revolution (*φ* = 2π) when the distance change *dL* is equal to the acoustic wavelength *λ*. From the wavelength and the frequency, we can calculate the speed of sound *u* which in turn depends on the environmental parameters; namely, temperature, humidity and pressure. The relationship between *u* and the environmental parameters is described by Cramer [20] and by Zuckerwar [21]. Thus, by measuring humidity and pressure and applying the inverted Cramer (or Zuckerwar) formula, it is possible to calculate *T* from *u*. The accuracy of the temperature estimation depends on many factors, the most relevant amongst which is the accuracy of the formulas. The overall uncertainty in the formula given by Cramer is 300 ppm, while the one claimed by Zuckerwar is 1000 ppm. This limits the accuracy of the estimation of temperature. A recent work carried out at INRIM was devoted to the experimental verification of the two formulas and demonstrated that, in the interval from 0 °C to 30 °C, it is possible to measure *u* with an uncertainty of 100 ppm leading to an uncertainty on *T* of 0.06 °C (results will be published soon).

## 3. The Instrument

A simplified schematic of the CAT is presented in Figure 1. The instrument is based on two piezo transducers (model HT 300PLT from HTW, China) able to generate and receive acoustic waves at 300 kHz. The choice of the frequency was driven by the consideration that, for a given displacement *L*, the phase change is proportional to the frequency (see formulas (2) and (3)), so high frequencies are preferable for achieving higher resolution. On the other hand, high frequencies are strongly attenuated by relaxation effects (at 300 kHz the attenuation exceeds 1000 dB per 100 m [22]). Finally, it is not easy to find ultrasound generators off the shelf designed to operate in air above 300 kHz. One of the two piezo transducers (the transmitter, Tx) is mounted on a translation stage fixed to a steel plate, while the second one (the receiver Rx) is fixed to the same plate. The moving stage is made of a microstep motor coupled to a precision screw which moves a carriage with a resolution of 0.12 µm over 100 mm (Zaber X-LHM100A). The stage is fixed to the steel plate so that the movement of the carriage is parallel to the axis between the centres of the two piezo transducers. A laser pointer fixed next to Rx is used to align the system. The distance between Tx and Rx varies from 300 mm to 400 mm. In principle, the distance can be substantially reduced, leading to a more compact realization; nevertheless, in this phase, we decided to maintain a minimum distance between the two piezo transducers in order to be sure to avoid cavity effects. The movement of the carriage was calibrated with an interferometer (Renishaw XL-80) in order to correct for the errors in the relative position of the stage with respect to the receiver, with an uncertainty of a few micrometres. The stage movement and position depend on temperature; thus, to guarantee the uncertainty level, the temperature of the translation stage must be monitored.

A simplified schematic of the electronic set-up is shown in Figure 2.

A dual channel synthesizer generates two signals at 300 and 310 kHz, respectively. The first one is used to excite the Tx to generate the ultrasonic wave. The wave is collected by the Rx and converted into an electric signal that is amplified by a low noise amplifier. The second signal is used together with two mixers to lower the operating frequency down to 10 kHz. The RF port of the first mixer is fed with a portion of the signal at 300 kHz generated by the synthesizer. The RF port of the second mixer is fed with the signal coming from Rx. The LO port of both mixers is fed with the signal at 310 kHz. At the LF port of the mixers, a 10 kHz signal is generated, filtered by a 270 kHz RC low-pass filter, and finally sent to the acquisition system. The two signals are sampled by a 16-bit acquisition board (NI PCI-6132) and sent to LabView-based software (SW). Here, the signals are further filtered with band-pass filters (two-pole Bessel filter, 4 kHz wide) and sent to a phasemeter. The output of the phasemeter is low-pass filtered (two-pole Bessel filter at 100 Hz). The band-pass and the low-pass filters are used to reduce the noise present on the weak Rx signal and improve the performances of the phase measurement; indeed, the piezo transducers are designed to be used at much shorter distance (a few mm rather than 400 mm), so the received signal is in the sub-millivolt range. The filtered output of the phasemeter is eventually recorded at a rate of 10 points per second. At the same time, the SW manages the motor driver in order to move the carriage at a constant speed of 5 mm/s. Speed fluctuations are minimum because of the high resolution and mechanical quality of the translation stage. A single measurement run lasts 20 s: in this time interval, the carriage is moved from one end to the other and the phase is measured continuously. At the end of the movement, the measured phase is plotted versus the travelled path, showing a linear behaviour due to the constant movement of the carriage. Then, the slope of the phase vs. displacement is calculated. As shown in Equation (3), this number is inversely proportional to the speed of sound in air. Therefore, for each measurement run, we obtain a value of the speed of sound. From this and from the measured values of ambient pressure and relative humidity, by applying the Zuckerwar model, one value of air temperature is calculated and recorded for each ramp.

In order to compare the CAT performances with traditional air temperature measurements, the SW acquires and records the temperature measured by several platinum resistance thermometers placed next to the CAT for each measurement run.

A screenshot of the LabVIEW software is shown in Figure 3 and a picture of the CAT is shown in Figure 4.

## 4. The Measurement Set-Up

The experiment was carried out at the INRIM laboratory for “applied thermodynamics and thermal metrology for climate” during the summer of 2021. A Kambic 105 climatic chamber with an extended range option, able to cover the whole range of atmospheric temperature and humidity values, was used to test the CAT against 4 reference platinum resistance thermometers (Pt100) connected to a Fluke 1586A super-DAQ readout system. Three thermometers were used to measure the air temperature in proximity of the CAT measurement volume, and the fourth was used to record the temperature of the moving carriage. A picture of the experimental set-up is shown in Figure 5. Measurements were performed in a temperature range from −30 °C to +55 °C and relative humidity ranging from 10%rh to 90%rh. In all conditions, the measurement of the CAT and of the Pt100 thermometers were recorded for about 1 hour, meaning that hundreds of data points were gathered for each experimental condition. The average of three Pt100 thermometers was evaluated and considered as the reference temperature of the climatic chamber to be compared with temperature values measured by the CAT. A typical measurement series is shown in Figure 6.

A second experiment was performed to demonstrate the immunity of the CAT to irradiation. The scheme of the setup and a picture are shown in Figure 7. A high-power halogen 1000 W projector was used to shine light through the glass of the chamber’s door. The whole CAT was illuminated, while the thermometers were illuminated in different ways depending on their position in the chamber. During the measurements, the chamber was kept at a constant temperature, and the light projector was switched on and off and moved in different positions.

## 5. Results and Discussion

A summary of the results obtained in the experiment is presented in Figure 8 in terms of CAT errors, i.e., differences between the temperature measured by the CAT and the reference temperature, calculated as the average of the three platinum thermometers. The lowest temperature achieved was −30 °C, limited by the freezing of the lubricant of the motorized actuator screw. The highest temperature was chosen to include the higher values of air temperature recorded near Earth’s surface. The black crosses represent the CAT errors obtained by averaging measurements registered for about one hour at that specific environmental condition. The red circles represent the average errors calculated for each temperature represented by the black crosses. The vertical red bar is the standard deviation of the measurements. The agreement between CAT and the traditional platinum resistance thermometers was well within ± 0.2 °C in the range −30 °C to +30 °C. At higher temperatures, the agreement worsened, although this was limited to 0.6 °C. This tendency must be investigated.

For each temperature, different values of RH were set in a range between 10%rh to 90%rh. In Figure 9, the distribution of the error of all the measurements with respect to RH is plotted in order to find a possible correlation between the two parameters. Although an excess error is observed at RH = 90%rh, a correlation was not observed, showing the independence of the CAT from RH.

In Figure 10, a typical result of the experiment implemented to demonstrate the difference between classical contact thermometry and acoustic thermometry in their dependence on irradiation is reported. In a single run of about five hours, four different irradiation conditions were generated. One was close to zero irradiation (the condition of all the measurements described so far); the others were generated by positioning the light projector in three different positions (distance and angle) in order to expose the three Pt100s and the CAT to different illumination conditions. It is evident that the heating of the Pt100s directly depends on the strength of the light, while, as expected, the CAT is immune to it. A thermometer in a solar shield was used as a reference for comparison (see picture in Figure 5).

Finally, we must consider the possible errors induced by environmental parameters such as the air movement (wind) and air pressure changes.

An air flow directed towards the CAT along a specific direction would affect the measured speed of sound and consequently the estimated temperature. A mathematical model to describe this effect can be found in [11]. The worst case occurs when the air flow is parallel to the measurement axis of the CAT; indeed, in this case, the measured speed of sound would be overestimated or underestimated by the same speed of wind. As an example, 1 m/s wind can cause an error up to 1.5 °C.

In our set-up, the climatic chamber fan caused a turbulent air movement in the measuring volume; hence, the measurement noise increased, but it did not introduce a systematic error in the measurements since the average air speed was zero. This explains the increased noise observed in the grey track in Figure 10 compared to the blue track in Figure 6: indeed, this is because, in the irradiance experiment, the fan was set to a maximum speed in order to maximize the temperature uniformity despite the high irradiance flux.

In order to overcome this issue in an open environment, a windshield could be used, or a second CAT could be arranged in the opposite direction, so that the error caused by parallel air flows can be removed by averaging the speed of sound measured by the two systems. 

The influence of air pressure changes can be considered negligible. Indeed, as seen in [20], a pressure change of 1 kPa corresponds to about 0.006 m/s change in the speed of sound (at 20 °C and 50%rh), i.e., a temperature change of 0.01 °C, which is well below the accuracy of the system.

## 6. Conclusions

We have presented a simple and compact instrument capable of measuring the thermodynamic temperature of air on a wide range of temperatures (T = −30 ÷ +55 °C) and relative humidity values (RH = 10 ÷ 90%rh). The CAT demonstrated an uncertainty of 0.2 °C in the interval −30 °C + 30 °C, increasing to 0.6 °C up to 55 °C. With some changes in the structure, the interval can be extended to higher and lower temperatures in order to include the Earth’s near-surface atmospheric temperatures and extreme conditions.

The main advantages of the CAT are that it is immune to irradiation errors and that a temperature calibration is not needed, since the traceability of the measurement is given by a mechanical measurement, exploiting its long-term stability. Furthermore, because of the absence of thermal inertia, it has a faster response than contact thermometers.

Further work will be devoted to the increase of the technological readiness level of the device, its compactness, and ease of use. The investigation will be extended to wider temperature and humidity values, with a focus on temperatures below 0 °C, in order to: (a) check potential condensation and icing effects and (b) cover the whole WMO prescribed range for temperature measurements ranging from −80 °C and + 60 °C.

## Figures and Tables

**Figure 1 sensors-23-01908-f001:**
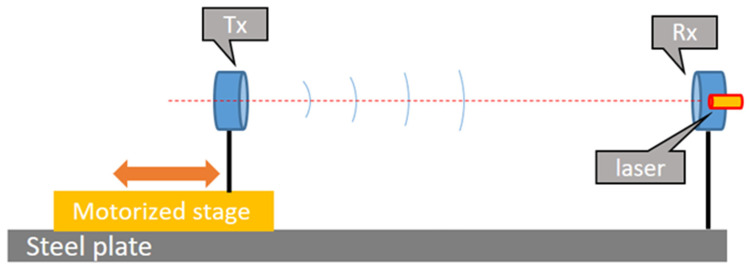
Simplified schematic of the CAT. Details are given in the text.

**Figure 2 sensors-23-01908-f002:**
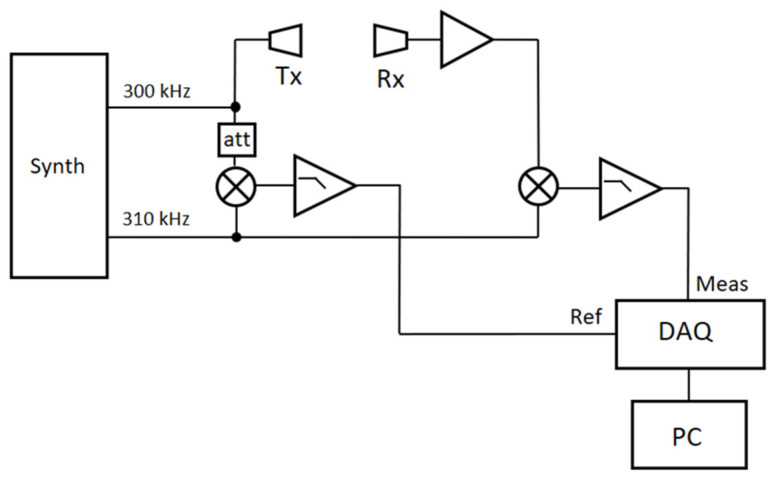
Simplified schematic of the electronics. See text for a detailed description of the components.

**Figure 3 sensors-23-01908-f003:**
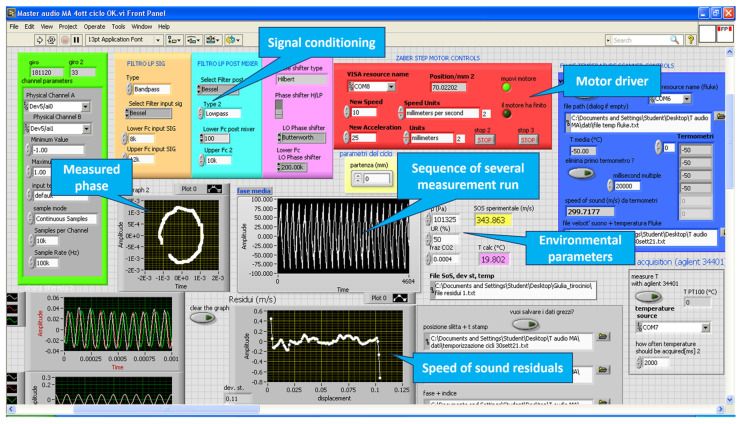
Screenshot of the LabVIEW software. Some of the key functions are indicated in the callouts. For each cycle, the measurement of the phase is recorded and the slope of the function phase/displacement is calculated. The lower graph shows the residuals of the function with respect to the linear fit. This gives an indication of the quality of each single measurement.

**Figure 4 sensors-23-01908-f004:**
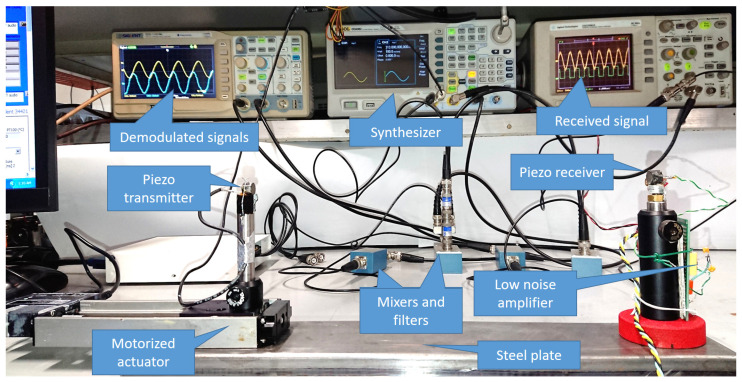
A picture of the CAT. In the foreground is the CAT unit mounted on the steel plate with the transmitter mounted on the translation stage (**left**) and the receiver fixed to the base (**right**). The amplifier board is mounted at the back of the holder of the receiver. On the table behind the CAT are the blue boxes with the mixers and the LP filters. The white box on the left is the connection to an A/D converter. On the bench are the synthesizer (**centre**) and the oscilloscope with the relevant signals: the RF amplified signal (yellow, (**right scope**)) and the LF demodulated and filtered signals (**left scope**).

**Figure 5 sensors-23-01908-f005:**
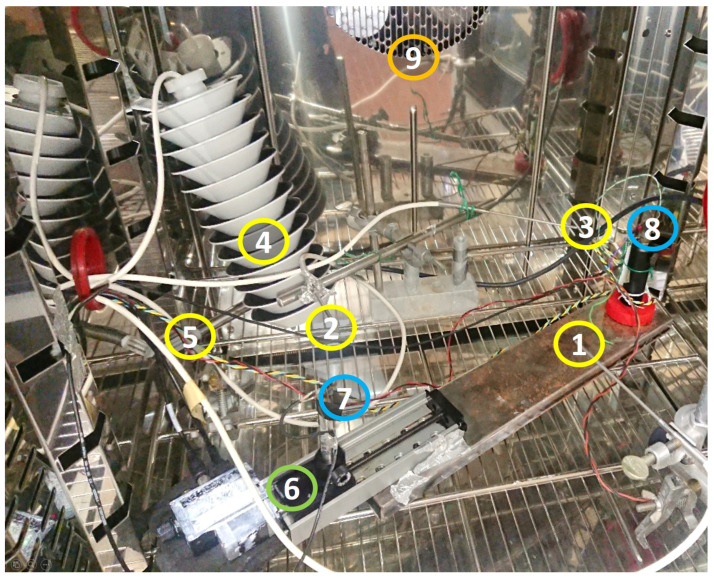
Picture of the experimental set-up. Points 1–4 indicate the reference platinum thermometers. Only in the case of the irradiation experiment is the thermometer marked as 4 inserted in a radiation shield as in the picture. Point 5 indicates the reference hygrometer. Point 6 is the motorized carriage. Point 7 is the piezo transmitter and 8 is the piezo receiver with the alignment laser on. Point 9 is the aperture with the fan from where the conditioned air comes out.

**Figure 6 sensors-23-01908-f006:**
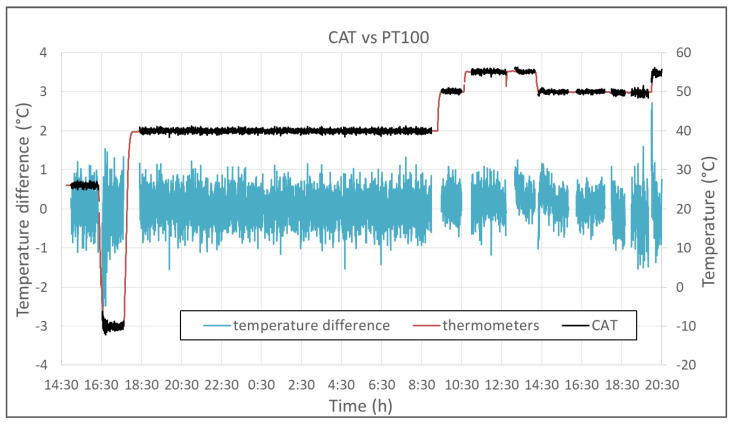
Typical measurement series. The red curve is the average of the temperatures measured by the three reference thermometers and considered as the reference temperature of the chamber (**right scale**). The temperature was set, respectively, to 26, −10, 40, 50, 55, 50, and 55 °C. In black: the temperature estimated from the speed of sound. The blue curve is the difference between the two curves (**left scale**). The interruptions in the data sets that can be observed at 55 and 50 °C refer to when the relative humidity set point was changed and the system was stabilizing to the new level.

**Figure 7 sensors-23-01908-f007:**
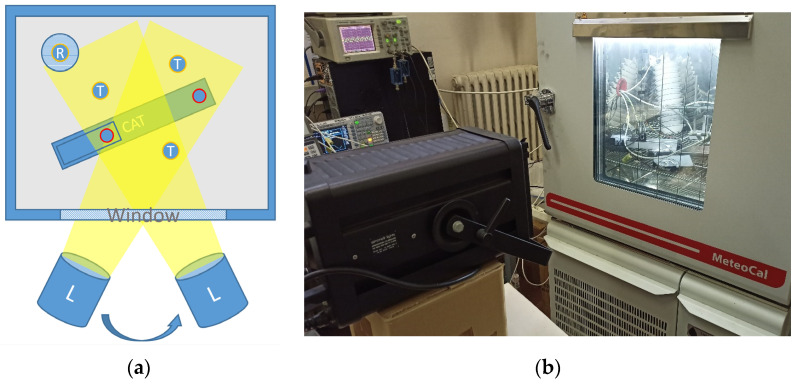
(**a**): Scheme of the set-up to check for the effect of irradiation where the lamp (L) is moved in different positions in front of the window. The marks T in the picture indicate the three naked Pt100, while the mark R stands for the radiation-shielded Pt100. The red circles represent the transmitter and the receiver of CAT. (**b**): A picture of the setup where the 1000 W halogen projector is visible in the foreground. On the right, the climatic chamber where the CAT and the thermometers are visible through the window.

**Figure 8 sensors-23-01908-f008:**
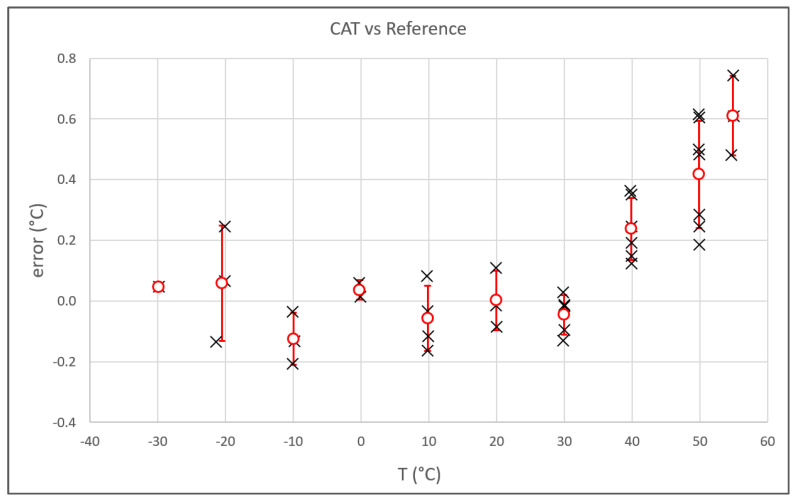
Summary of the results: the measurement difference between the CAT and the reference thermometers is plotted vs. the temperature. The red circles represent the average of the results for each temperature. The vertical red bars are the standard deviation of the differences.

**Figure 9 sensors-23-01908-f009:**
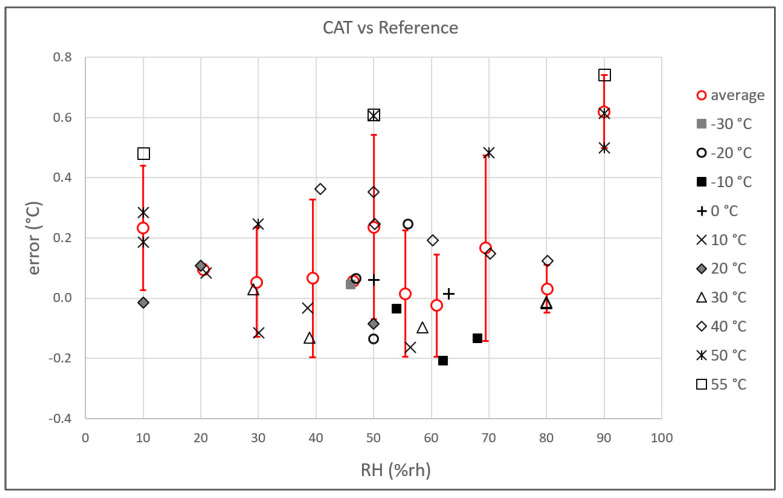
Summary of the results: the measurement difference between the CAT and the reference thermometers is plotted vs. the relative humidity. The temperature of each individual measurement is indicated in the legend. The red circles represent the average of the results grouped for each relative humidity value. The vertical red bar is the standard deviation of the measurements.

**Figure 10 sensors-23-01908-f010:**
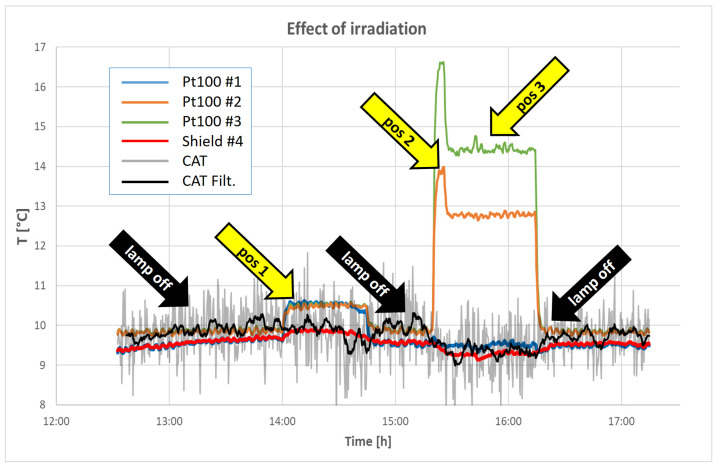
Typical result from the experiment to check the effect of irradiation. The set-up of the experiment is explained in the text. The blue, orange, and green lines are the temperatures recorded by three platinum thermometers used to calculate the reference temperatures for the previous measurements. The red line reports the records of a reference thermometer equipped with a solar shield screen. The labels #1 to #4 refer to the picture in Figure 5. The light grey line is the raw temperature measured with the CAT, while the black line is the moving average of the CAT readings during a period of 5 min. The yellow arrows (positions 1 to 3) indicate where the lamp is switched on and where its position is changed.

## Data Availability

Relevant data used to produce the results published in this work are available at: https://doi.org/10.5281/zenodo.7611072.

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
