# Peer review of "Non-Contact Thermometer for Improved Air Temperature Measurements"

_sensors, 2023, doi:10.3390/s23041908_

Round 1
Reviewer 1 Report
I have reviewed the manuscript titled "Non-contact thermometer for Improved Air Temperature Measurements" and have found it to be a valuable contribution to the field. The authors have developed a compact thermometer for measuring the temperature of air based on the measurement of the speed of sound and have compared it to platinum resistance thermometers in a climatic chamber. The instrument demonstrated an overall measurement uncertainty of 0.2 °C in the range from -30 °C to +30 °C, was immune to irradiation errors, and did not require a temperature calibration.
While the manuscript is generally well written and the results are clearly presented, there are a few minor corrections that should be addressed before publication.
The bibliography needs to be extended and the Introductory section updated accordingly by including references to temperature sensors from the state-of-the-art (resistive, capacitive, optical, microwave, etc.).
In the abstract, the authors present a thermometer for air temperature, but the measurement range presented here is -30 °C to +30 °C, which only partially covers the ambient temperature range. To better highlight the performance of the proposed device, I would suggest reporting in the abstract the full temperature range investigated in this study (i.e., from -30 °C to 55 °C) as well as the range in which the measurement uncertainty is 0.2 °C.
After the signals have been acquired using the NI PCI-6132 board, they have been filtered. However, it is not clear which type of digital filter was applied (Butterworth, Chebyshev filter, ...).
The readability of the plots in Figures 6, 8, 9, and 10 could be improved. Legends should be included and the same notation should be used for measurement units ("()" or "[]"). In Figure 6, it is not clear which axis should be considered for the plotted data. In Figure 9, "UR (%)" should be replaced with "RH (%rh)." In Figure 10, it is not immediately clear the meaning of the yellow arrows.
There are a few typos throughout the text that should be corrected. For example, in line 112, "t" should be deleted and "T" and "u" should be written in italics. In line 183, "picture 5" should be changed to "Figure 5." In line 219, a period is likely missing after "Earth's surface."
Finally, I would suggest writing "Pt100" instead of "PT100" and using %rh as the measurement unit for relative humidity (RH) (e.g., "and of relative humidity ranging from 10 %rh to 90 %rh.")
Author Response
We thank the reviewer for the critics and the suggestions that allowed ut to improved the paper. We believe we have addressed all the points. A detailed list of the actions can be found in the attachment.

Reviewer 2 Report
The reference list must be improved. It is too poor.
Your proposed device must be better compared to the exhisting techniques supporting this with adequate references
The influence of wind, air movement and pressure changes must be better analyzed
Author Response

(The authors gave the same response as above.)

Reviewer 3 Report
sensors-2162072 Non-contact thermometer for improved air temperature measurements
I find the article an interesting contribution and I think it can be published in SENSORS. However, I have some comments for the author.
Some comments have to be added about the chosen value of the frequency
Some comments have to be added about the spectra of Tx and Rx
Line 201 please verify temperature values
Fig.9 horizontal axis: not UR but RH
Two plots have to be added, with error (°C) as a function of RH% similar to fig9, one for T=20°C and other for T= 26°C
Fig 10: in the caption, the references to fig 5 for the PT100 positions have to be added, together with some comments on blue line behaviour
Fig 10: black line shows a less stable value over time than the value of the shielded and not shielded PT100s: some comments have to be added
Summarizing, I think that the article can be published in SENSORS, after the required corrections have been completely made.
Author Response

(The authors gave the same response as above.)

Round 2
Reviewer 2 Report
for the paper is fine in its actual form
Author Response
Thank you very much for noticing it, indeed it was wrong. In order to correct the error, we have replaced "accuracy" with "uncertainty" at the beginning of the sentence.